# ZnO/Graphene Composite from Solvent-Exfoliated Few-Layer Graphene Nanosheets for Photocatalytic Dye Degradation under Sunlight Irradiation

**DOI:** 10.3390/mi14010189

**Published:** 2023-01-12

**Authors:** Vasanthi Venkidusamy, Sivanantham Nallusamy, Gopalakrishnan Nammalvar, Ramakrishnan Veerabahu, Arun Thirumurugan, Chidhambaram Natarajan, Shanmuga Sundar Dhanabalan, Durga Prasad Pabba, Carolina Venegas Abarzúa, Sathish-Kumar Kamaraj

**Affiliations:** 1Department of Physics, National Institute of Technology-Tiruchirappalli, Tiruchirappalli 620015, India; 2Department of Physics, K. Ramakrishnan College of Engineering, Tiruchirappalli 621112, India; 3School of Physics, Madurai Kamaraj University, Madurai 625021, India; 4Sede Vallenar, Universidad de Atacama, Costanera #105, Vallenar 1612178, Chile; 5Department of Physics, Rajah Serfoji Government College (Autonomous), Thanjavur 613005, India; 6Functional Materials and Microsystems Research Group, RMIT University, Melbourne, VIC 3000, Australia; 7Departamento de Ingeniería Mecánica, Facultad de Ingeniería, Universidad Tecnologica Metropolitana, Santiago 8330378, Chile; 8Centro de Investigación en Ciencia Aplicada y Tecnología Avanzada (CICATA)-Unidad Altamira, Instituto Politécnico Nacional (IPN), Carretera Tampico-Puerto Industrial Altamira Km 14.5, C. Manzano, Industrial Altamira, Altamira 89600, Mexico

**Keywords:** ZnO/graphene composite, liquid phase exfoliation, photocatalytic activity, methylene blue, 1,2-dichloroethane

## Abstract

ZnO/graphene nanocomposites were prepared using a facile approach. Graphene nanosheets were prepared by ultrasonication-based liquid phase exfoliation of graphite powder in a low boiling point organic solvent, 1,2-Dichloroethane, for the preparation of ZnO/graphene nanocomposites. Structural properties of the synthesized ZnO/graphene nanocomposites were studied through powder XRD and micro-Raman analysis. The characteristic Raman active modes of ZnO and graphene present in the micro-Raman spectra ensured the formation of ZnO/graphene nanocomposite and it is inferred that the graphene sheets in the composites were few layers in nature. Increasing the concentration of graphene influenced the surface morphology of the ZnO nanoparticles and a flower shape ZnO was formed on the graphene nanosheets of the composite with high graphene concentration. The efficiencies of the samples for the photocatalytic degradation of Methylene Blue dye under sunlight irradiation were investigated and 97% degradation efficiency was observed. The stability of the nanocomposites was evaluated by performing five cycles, and 92% degradation efficiency was maintained. The observed results were compared with that of ZnO/graphene composite derived from other methods.

## 1. Introduction

Graphene, one of the attractive members of the carbon family, is a vastly investigated material of this century [1,2]. The mechanical, thermal, electrical, and physicochemical qualities that are unique to graphene and metal/metal oxide composites made with graphene have led to innovative developments in the fields of energy generation, energy storage, water purification, water splitting, sensors, etc [3,4]. In recent decades, the rapid rise of industries and population has led to water pollution and hence water treatment is an important issue to be resolved [5,6,7,8,9,10]. Various metal oxides like TiO_2_, ZnO, Cu_2_O, SnO_2_, MnO_2_, ZnFe_2_O_4_, CuFe_2_O_4,_ and Bi_2_WO_6,_ etc., were combined with the graphene and used as an adsorbent for environmental applications [11,12,13,14]. The utilization of graphene for nanocomposite formation is due to the unique physicochemical characteristics of the graphene which could enhance the physicochemical properties of the other materials in the nanocomposites and the synergic effect could be utilized in various applications [15,16]. ZnO is the most significant choice in a sustainable environment among several semiconductor photocatalyst substances due to its great oxidation capability, thermal and chemical stability, non-toxicity, good photocatalytic and photosensitive capabilities, and biocompatibility [17]. An enhanced surface area and concentration of active sites in ZnO nanoparticles with smaller size help in the absorption of organic molecules and enable the reaction of photogenerated charge carriers with pollutant molecules [3,18,19]. Due to the superior performance of graphene composites compared to individual oxide nanomaterials, there has been an increase in efficiency in a variety of fields, including energy conversion, energy harvesting/storage devices, optoelectronic devices, and photocatalysis [20,21]. Mostly, the composite of graphene/metal oxide was prepared from the reduced graphene oxide solution derived by modified Hummer’s method, electro-chemical and thermal methods [22], and many efforts like chemical, thermal, electrochemical, and microwave reduction methods were introduced to remove the functional groups present in the exfoliated graphene oxide [23,24,25], which affects the electron transport properties of GO. However, hazardous reducing chemicals like hydrazine or high-temperature processes were used to reduce graphene oxide [26], and the reduced graphene oxide solutions were directly used in the solution process to produce graphene/metal oxide nanocomposite [27].

Nanocomposites of GO/ZnO and rGO/ZnO are highly efficient and widely studied candidates in the photocatalytic degradation of industrial Methylene Blue (MB) dye, and various methods have been adopted to synthesize ZnO/graphene nanocomposite materials [28,29,30]. However, the basal plane defects and reduction of band gap affect the photocatalytic efficiency of rGO/ZnO. The graphene sheets prepared from shear-assisted or ultrasonication-based liquid phase exfoliation have low edge defects and are without basal/sp^3^ defects [31,32]. Hence, the composite of graphene/ZnO may lead to improved photocatalytic degradation efficiency. Due to the high solubility of MB in water and high toxicity, the degradation of MB is crucial to avoid environmental pollution. Recently, Rouzafzay et al. reported the photocatalytic activity of graphene/ZnO nanocomposite synthesized from shear-exfoliated graphene sheets and achieved a degradation of 99% against MB after irradiation of 180 min under visible light [30].

Here, we have reported a facile technique for the development of ZnO/graphene nanocomposite using few-layer graphene sheets produced by ultrasonic exfoliation of graphite in 1,2-Dichloroethane. To our knowledge, there are no available reports of graphene/ZnO nanocomposite prepared using ultrasonic-based liquid phase exfoliated graphene nanosheets. Further, the structural, morphological, and optical properties of the synthesized graphene/ZnO nanocomposites were studied. Additionally, the produced composites’ photocatalytic activity was examined in relation to MB dye degradation when exposed to sunlight.

## 2. Materials and Methods

(i) Preparation of Graphene

Graphite powder of nearly 0.15 g purchased from Sigma Aldrich was added into the 50 mL of 1,2-Dichloroethane taken in measuring flasks. Then, the graphene and solvent mix was sonicated for 5 h by means of an ultrasonic bath. The sonicated dispersions were centrifuged at 3000 rpm for up to 30 min. After that, the supernatant dispersions were collected, and the solvent was evaporated at room temperature for nearly 48 h to obtain the exfoliated graphene nanosheets.

(ii) Preparation of ZnO/Graphene composite

Zinc nitrate hexahydrate dissolved in double distilled water was added to the exfoliated graphene dried from 85 mL of graphene dispersion. The solution of NaOH was added to increase the pH of the graphene/zinc nitrate hexahydrate dispersion to 12. Then the solution mixer was heated at 70 °C for 2 h by maintaining the pH of the solution. The obtained precipitate was washed with distilled water and dried at 100 °C. Then, the final particles were heated at 400 °C for 2 h. The same procedure was carried out with graphene sheets dried from 175 mL of graphene dispersion. The ZnO/graphene nanocomposite prepared from 85 and 175 mL of graphene dispersion are named S1 and S2, respectively.

(iii) Characterizations

Structural properties of the ZnO/graphene nanocomposites were studied using powder XRD pattern recorded for the 2θ range of 20° to 80° with a step size of 0.01° using PANanalytical Empyrean with Cu-K_α_ radiation and room temperature micro-Raman spectra were measured using HORIBA Jobin Yvon LabRAM HR 800 with He-Ne laser of wavelength 632.81 nm as an excitation source. FEI NOVA NANOSEM 450 was used to study the surface morphology of the samples. The diffused reflectance spectrum of the powder samples and UV-Vis absorption spectra of the collected samples after irradiation with nano photocatalyst were recorded using a SHIMADZU UV-2450 spectrophotometer.

(iv) Photocatalytic experiments

The photocatalytic property of the synthesized ZnO/graphene nanocomposites was studied by examining the photodegradation of MB dye under sunlight irradiation. Nearly 50 mg of ZnO/graphene nanocomposites were added into 150 mL aqueous solution of MB with 1 × 10^−5^ M concentration. Then, the dispersion with the nano photocatalyst was stirred in a dark room for 30 min. Further, the solution was exposed to sunlight irradiation and 2 mL of sample solution was collected after a regular interval of 20 min. The collected samples were centrifuged at 3000 rpm speed. The concentration of MB in the supernatant of centrifuged samples was estimated from the absorption spectra of MB. The photocatalysis experiment was performed on a sunny day during summer (April) between 12:00 PM to 3.30 PM (average solar radiation is ~20 MJ/m^2^) at National Institute of Technology Tiruchirappalli, Tamilnadu, India in an atmospheric environment.

## 3. Results and Discussion

The XRD patterns of the ZnO/graphene nanocomposite with different concentrations of graphene are given in Figure 1. The observed peaks are belonging to the hexagonal wurtzite crystal structure of ZnO [JCPDS No. 36–1451] [33] and the estimated lattice constant for the sample S1 is a = b = 3.245 Å, c = 5.199 Å and for the sample, S2 is a = b= 3.240 Å, c = 5.192 Å. The diffraction peak of graphene/graphene oxide is not traced in both of the samples. The absence of the diffraction peak of graphene may be due to the lesser concentration of graphene in the samples. The crystallite size of the samples S1 and S2 estimated using the Scherrer formula [34,35] is about 35 and 84 nm, respectively. This increase in crystallite size of S2 compared to S1 implies the influence of graphene concentration in the growth of ZnO particles [36,37].

Figure 2 shows the micro-Raman spectra of samples S1 and S2. The micro-Raman spectra of exfoliated graphene nanosheets (EGS) and bulk graphite powder are shown in Appendix A for comparison. The observed Raman bands of the samples S1 and S2 around 100, 200, 319, 425, 558, 644 and 1095 cm^−1^ are corresponding to E_2L_, 2E_2L_, E_2H_-E_1L_, E_2H_, A_1L_/E_1L_, 2E_2H_-E_2L,_ and A_1_ + E_2L_ modes, respectively, of hexagonal wurtzite ZnO structure [38]. The Raman band at 558 cm^−1^ belongs to A_1L_/E_1L_ mode which is related to zinc interstitial and oxygen vacancy defects in the ZnO lattice [39,40].

Along with the Raman active modes of ZnO, the Raman bands of graphene are also observed at ~1318 cm^−1^ (D band), ~1567 cm^−1^ (G band), 1608 cm^−1^ (D’ band), and ~2639 cm^−1^ (2D band) for S1, and at ~1322 cm^−1^ (D band), ~1563 cm^−1^ (G band), 1595.7 cm^−1^ (D’ band), and ~2641 cm^−1^ (2D band) for S2. The D and G band positions and the intensities were slightly changed for S2 when it is compared to S1 nanocomposites, which might be due to the higher fraction of graphene in the ZnO/graphene nanocomposites (S2). The intensity ratio of 2D and G band (I_2D_/I_G_) of graphene bands of S1 and S2 is 0.53 and 0.30, respectively, indicates the few-layer nature of the graphene nanosheets in both composites. The number of graphene nanolayers per flake of graphene is calculated from the Raman shift of the G band using the following relation [30],
(1)$$\omega_{G}=1581.7+\frac{11}{1+n^{1.7}}\;\;\;\;\;\;\;\;\;\;\;\;\;\;\;\;\;\;\;\;\;$$
where $\omega_{G}$ is the Raman shift of the G band and n is the number of layers per flake. Another relation found by Paton et.al [31] to calculate the number of graphene layers per flake is
(2)$$N_{G}=10^{\left(0.85M+0.45M^{2}\right)}$$
where
M=Igraphene (at ω=ωp, graphite)Igraphene (at ω=ωs, graphite)Igraphite (at ω=ωp, graphite)Igraphite (at ω=ωs, graphite)

The calculated values of the number of graphene layers using the relations (1) and (2) are given in Table 1, and it inferred that the graphene nanosheets in both samples S1 and S2 are few-layer graphene.

The D band gives information on the edge and topological defects and the ratio of the intensity of the D to G band (ID/IG) provides the details of the defect population. The value of ID/IG for samples S1 and S2 are 0.54 and 0.40, respectively, greater than that of exfoliated graphene sheets (ID/IG = 0.3) (Appendix A) and bulk graphite (ID/IG = 0.07). As the ratio of ID/IG is indicating the structural disorder and size of the sp2 domain of the graphene nanosheets, the increased ID/IG of ZnO/graphene nanocomposites compared to the exfoliated graphene sheets implies the interaction of ZnO with the edge defects of the graphene nanosheets during the composite formation [41]. Similarly, from the ratio of the intensity of the D band to the D’ band (ID/ID’), we can characterize the nature of the defect present in the graphene. The value of ID/ID’ is the minimum for the edge/boundary defects (~3.5) in graphite powder and it increases to ~7 for the graphene nanosheets with basal/vacancy defects [30,31]. It reaches a maximum of ~13 for the graphene nanosheets with sp3 defects. From the calculated ID/ID’ of samples S1 and S2, we can conclude that the defects in the present samples are purely edge defects and are free from basal defects.

As it is evident that the samples consist of only edge defects, we can calculate the approximate mean lateral size (L) of the flakes and crystallite size (La) from the following relations [42]
(3)$$\mathrm{Mean}\;\mathrm{lateral}\;\mathrm{size}\;L=\frac{0.17}{\frac{I_{D}}{I_{G}}\;\mathrm{of}\;\mathrm{Graphene}-\frac{I_{D}}{I_{G}}\;\mathrm{of}\;\mathrm{Graphite}\;}$$

Crystallite Size
(4)$$L_{a}\;\left(\mathrm{nm}\right)=\left(2.4\;\times \;10^{-10}\right)\lambda_{l}^{4}{\left(\frac{I_{D}}{I_{G}}\right)}^{-1}$$where $\lambda_{l\;}$ is the wavelength of the excitation laser line used in Raman spectroscopic measurement. The $\frac{I_{D}}{I_{G}}\;$ of graphite powder used is 0.08. The calculated mean lateral size and in-plane crystallite size of graphene in samples S1 and S2 are given in Table 1.

Well-dispersed graphene sheets with agglomerated particles of ZnO are clearly seen in the SEM micrograph of sample S1 shown in Figure 3a,b. The information derived from the SEM images of the exfoliated graphene nanosheets implies that there is no change in the dimension and morphology of graphene nanosheets after the incorporation of ZnO. The size of the graphene nanosheets was observed in a wide range of sizes ranging from a few hundred nm to a few tens of micrometers as observed in Appendix A. Highly crystalline ZnO micro-flowers with a size range from 0.5–1.5 μm formed on the surface of graphene sheets and are observed in the SEM images of S2 (Figure 3c,d). These results are indicating the influence of the increasing concentration of graphene in the further growth of ZnO in the S2 sample. As observed in the Raman result, the existing carbon defects in the graphene sheets have a high tendency to bind with the oxygen atoms of ZnO. The strong interface of ZnO with the defect sites of graphene nanosheets hindered the aggregation of ZnO particles and thus individual ZnO flowers are observed in the S2 sample [38]. Further, the high concentration of graphene sheets in the S2 sample enhanced the growth of ZnO, resulting in the increased particle size, and ended with flower-like morphology. The well-dispersed graphene nanosheets in the S1 sample enhanced the nucleation and restricted the growth of the ZnO, which is reflected in the particle size of ZnO in S1 with less graphene concentration [43].

The elemental compositions of the samples and functional groups/elements attached to the carbon atoms of graphene are studied using X-ray photoelectron spectroscopy. Figure 4a shows the XPS core level spectra of the Zn 2p in ZnO/graphene nanocomposites. The observed binding energy peaks at 1021.68 and 1045.02 eV are corresponding to the electron orbits of Zn 2p_3/2_ and Zn 2p_1/2_. The binding energy peaks at 531.1 and 533.5 eV in Figure 4b are corresponding to the O^2−^ in the oxygen-deficient regions in the ZnO matrix [44], and surface hydroxyl (-O-H) groups, respectively [45,46]. The XPS spectrum of C1s is shown in Figure 4c and the strong binding energy peaks at 285.0 eV and 288.4 eV are corresponding to the C-C bonding in sp^2^ hybridization and C-O bonding, respectively, whereas the weak binding energy peak at 289.6 eV is belonging to the O-C=O bonding [47]. The peak observed at 286.8 eV is assigned to the −C−O−Zn bond formed in the samples [48,49,50]. These results also ensured the formation of ZnO/graphene nanocomposite from the solvent-exfoliated graphene sheets.

Figure 5 shows the UV−Vis diffused reflectance spectra of the ZnO/graphene nanocomposites. The observed reflectance edge of the samples at 380 nm is corresponding to a band gap of 3.2 eV. The broad reflectance observed in the ultra-violet region may be attributed to the π-π* transition of the atomic C-C bonds [51]. The band gap variation of ZnO after composite formation with graphene was estimated using Kubelka–Munk (K-M) relation (hν(F(R))^2^ versus hν)[52] as shown in the insert of Figure 5. The obtained band gap of S1 and S2 samples decreases to 2.94 and 2.90 eV compared to that of pure ZnO (3.2 eV). This notable reduction of the band gap of the samples is due to the covalent band created between zinc oxide and graphene [53,54,55]. The slight variation in the band gap of the S2 sample compared to the S1 sample might be due to the particle size growth and also the higher fraction of graphene in the ZnO/graphene nanocomposites. This narrowing of the band gap of the nanocomposites enhances the degradation of visible light absorption. The UV-Vis spectra of bare graphene nanosheet dispersion are shown in the Appendix A.

The PL spectra of the samples excited at 330 nm are shown in Figure 6. The S1 and S2 samples show emission peaks at 415, 451, 468, 482, and 493 nm. The recombination of electrons from the conduction band to the oxygen interstitial (Oi) energy level is responsible for the prominent wide emission peak at 415 nm. The blue emission peaks at 451 and 468 nm are due to the electron transition between the conduction band minimum and Zn vacancy sites (V_zn_), and the transition between the zinc interstitials and valence band maximum [56,57]. The weak shoulder peaks at 482 and 493 nm may be attributed to the oxygen vacancies (V_o_) and free excitons (FX) that existed in the ZnO crystal lattice of the composite [58]. The observed visible emission bands indicated the existence of various defect energy levels in both of the samples. The drop in emission intensity of S1 implies better electron transfer between the ZnO and graphene nanosheets than that of sample S2, which may result in a reduction of electron-hole recombination and better photocatalytic activity of sample S1 [59].

The mechanism of ZnO/graphene’s photocatalytic activity on MB dye in sunlight is schematically depicted in Figure 7. In the dark, initially, the MB dye molecules are adsorbed on the surface of the ZnO/graphene nanocomposite by means of electrostatic interaction and π-π interaction with the carbon atoms of graphene. After sunlight irradiation, the electrons in the valence band (VB) of ZnO are excited to the conduction band (CB) and created an effective number of photo-generated electron-hole pairs. Then, the electrons in the CB of ZnO and the delocalized electrons in graphene react with the oxygen atoms in the dye solution and produce oxy radicals. Meanwhile, oxy radicals and holes in the VB react with the H_2_O molecules and produce hydroxyl radicals. These reactive hydroxyl radicals degraded the dye molecules into non-toxic molecules [60,61]. According to the PL spectra, the defect energy level in ZnO traps excited electrons and reduces the formation of electron-hole pairs, supporting improved dye degradation. Due to the slightly higher energy level (−4.05 eV) of the conduction band of ZnO than the work function of the graphene sheet (−4.42 eV), the electrons in the CB of ZnO readily transferred to the graphene and the electrons are involved in the dye degradation [62]. The presence of graphene nanosheets with ZnO increases the degradation efficiency as it improves the transportation and separation of photogenerated electron-hole pairs and their lifetime at the semiconductor/solution interface. Moreover, the high surface area of graphene nanosheets provides a desirable interfacial hybridization with ZnO. Hence, the overall dye degradation is expected to be improved after the incorporation of graphene in ZnO [63].

The UV-Vis absorption spectra of the MB solution after various periods of sunlight irradiation with S1 and S2 samples are shown in Figure 8a,b. In the case of sample S1, the adsorption of dye on the photocatalyst in the dark is not significant. However, under the irradiation of sunlight, nearly 75% of MB dye is degraded in 20 min. Corona et al. have reported 80% of degradation efficiency for the nitrogen-doped ZnO-graphene structure after 25 min irradiation under a visible light source [64]. Further irradiation increases the degradation efficiency and nearly 97.5% of MB is degraded after 180 min. This result is highly appreciated when compared to the results of 10 mg of ZnO/graphene photocatalyst prepared from rGO derived by modified Hummer’s method and electrolysis process [65,66]. The adsorption of dye under dark conditions is increased in the case of the S2 sample (Figure 8b) due to the higher concentration of graphene which enhanced the adsorption capacity of ZnO. Figure 8c shows the percentage of degradation efficiency (%) of the photocatalysts for the degradation of MB under sunlight at different intervals of irradiation time calculated using the following equation,
(5)$$\mathrm{Degradation}\;\mathrm{efficiency}\;\eta =\frac{C_{0}-C}{C_{0}}\times 100\%$$
where C_0_ represents the starting concentration of MB and C represents the concentration of MB at time t under the influence of sunlight. The degradation of MB with S2 is gradual and nearly 83.5% of MB was degraded after 180 min of sunlight irradiation. The sample S1 with low graphene concentration shows better degradation performance than S2. The poor performance of S2 compared to S1 is due to the formation of thicker graphene layers in S2. According to SEM pictures, the thicker graphene layers created on ZnO absorb a significant amount of incidental light and lowered ZnO photo-absorption, which lowers the effectiveness of S2 photocatalytic degradation [67].

However, the greater surface area and higher defect concentration of S1’s sample, which is smaller than S2’s, can be attributed to its improved degrading efficiency. The improved degrading performance of the S1 sample is also significantly influenced by the point defects in the ZnO. The ZnO lattice’s zinc and oxygen vacancies served as a focus for charge carriers to be trapped, delaying the recombination of electrons and holes. The better electron transfers of sample S1, as proved by less PL intensity compared to S2, may lead to its enhanced photocatalytic activity. The degradation efficiency of sample S1 on MB dye is comparable with that of ZnO/graphene, ZnO/GO and ZnO/rGO reported recently. Moreover, in the previously reported works, in Table 2, the highest rate of dye degradation within a short period has been achieved using high power ultra-violet and visible light lamps as the irradiation source. However, in the present work, photocatalytic activity was performed under sunlight irradiation. Hence, it is evident that the degradation efficiency of the sample S1 prepared from solvent-exfoliated graphene nanosheets is more reliable and the best alternative for ZnO/GO and ZnO/rGO composites as photocatalysts.

The following pseudo-first-order equation is used to study the photocatalytic degradation of MB for ZnO/graphene nanocomposites,
ln(C_0_/C) = kt (6)
where k -apparent reaction constant. The calculated rate constants for S1 and S2 samples from the slope of ln(C_0_/C) vs ‘t’ plots shown in Figure 8d are 0.0047 min^−1^ and 0.0034 min^−1^, respectively. This value is comparable with the degradation rate constant of rGO/ZnO nanocomposites for the photocatalysis of MB under UV light radiation [76,77].

The stability of the sample S1 with high degradation efficiency is studied by performing the photocatalytic test for five repeated cycles and it is shown in Figure 9. For each cycle, the photocatalyst was filtered, washed with double distilled water, and then used for many cycles. The efficiency of sample S1 on the degradation of MB was reduced to 92% after 180 min of sunlight irradiation. This indicates the high stability of the sample under the photocatalytic process and sunlight irradiation.

The present work has the following advantages, such as (i) the synthesis process is simple, cost and time-effective, and is also free from hazardous chemicals, (ii) the synthesized ZnO/Graphene composite contains few-layer graphene sheets without any basal plane defects, (iii) the solvent used for the graphene exfoliation gives a considerable yield of exfoliated graphene and it can be evaporated (low boiling point 84 °C) at room temperature, hence, the removal of solvent from the graphene is quite easy, which is not possible for other well-known solvents used for graphene exfoliation like, DMF, NMP, etc., and (iv) the graphene sheets prepared by the present method show a better quality than that of graphene sheets prepared from shear exfoliation.

## 4. Conclusions

In this work, a simple, inexpensive two-step solution processing technique was reported for the synthesis of ZnO/graphene nanocomposite. The results obtained from the XRD pattern, micro-Raman spectrum, SEM images, XPS spectrum, and UV-Visible spectrum have confirmed the successful preparation of ZnO/graphene composite from graphene nanosheets exfoliated in 1,2-Dichloroethane. The graphene sheets lack basal plane flaws, and the present edge defects make it easier for oxygen to connect to a carbon atom, forming the C-O-Zn binding complex. The results have also implied the influence of graphene concentration on the growth of ZnO on the surface of graphene nanosheets. The ZnO/graphene nanocomposite prepared with less graphene concentration showed a remarkable photocatalytic degradation efficiency of 75% within 10 min and 97% after 180 min of sunlight irradiation. These results indicated the successful preparation of ZnO/graphene nanocomposite from few-layer graphene sheets exfoliated in 1,2-Dichloroethane by ultrasonication and revealed an excellent performance in the photocatalytic degradation of MB dye under sunlight irradiation. The new synthesis method could be adopted for further synthesis of graphene materials.

## Figures and Tables

**Figure 1 micromachines-14-00189-f001:**
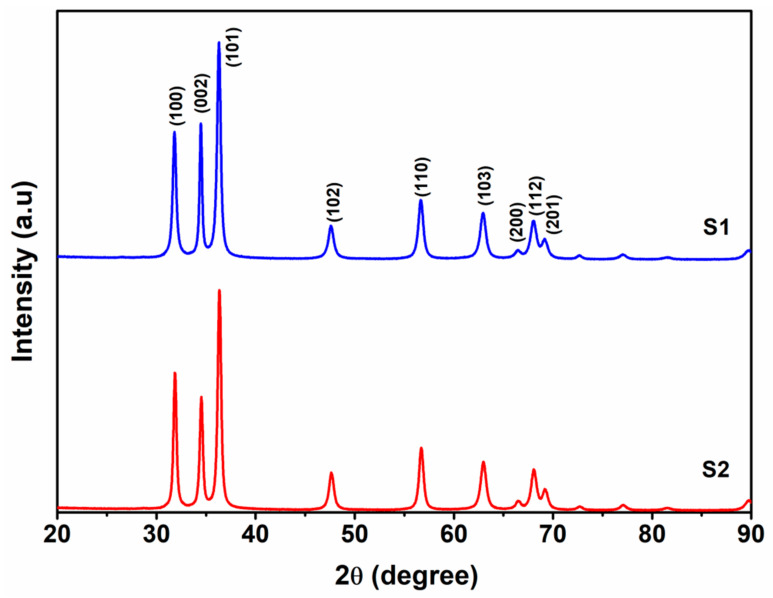
XRD patterns of ZnO/graphene nanocomposites.

**Figure 2 micromachines-14-00189-f002:**
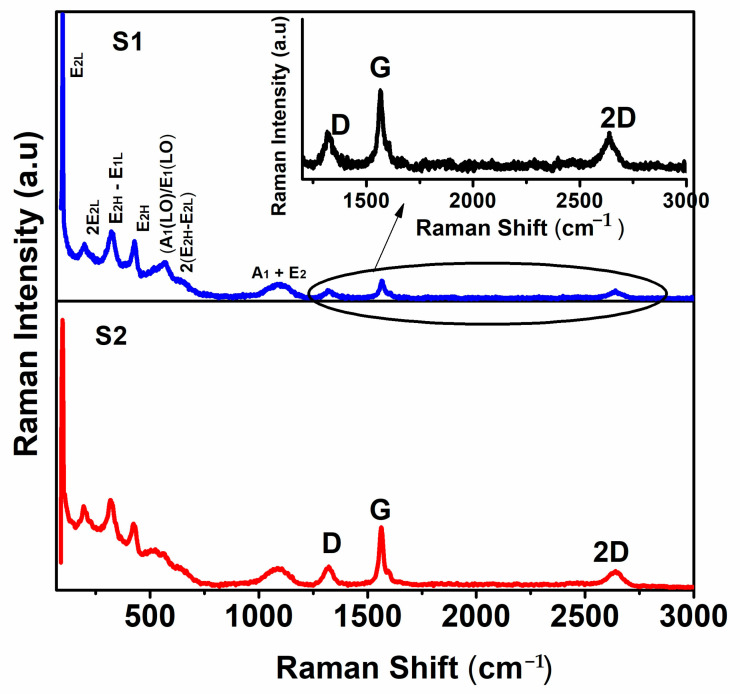
Micro Raman spectra of ZnO/graphene nanocomposites.

**Figure 3 micromachines-14-00189-f003:**
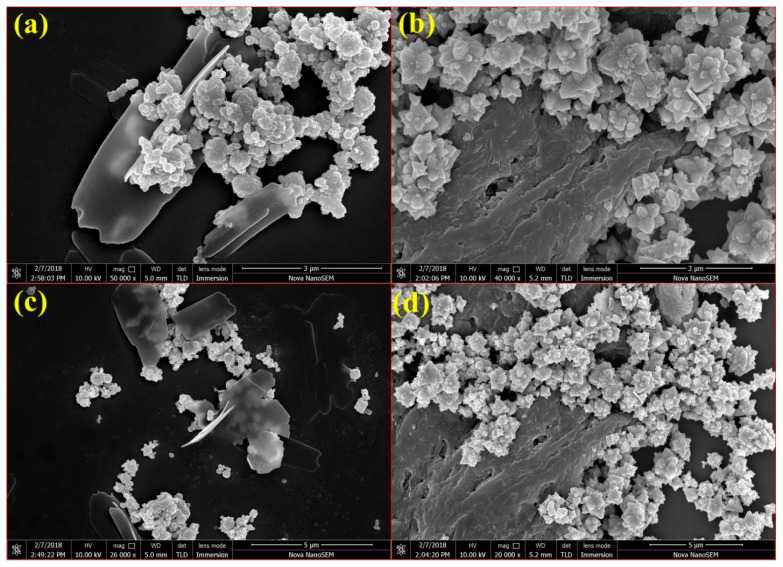
SEM micrograph of ZnO/graphene nanocomposites(**a**,**b**) S1, and (**c**,**d**) S2.

**Figure 4 micromachines-14-00189-f004:**
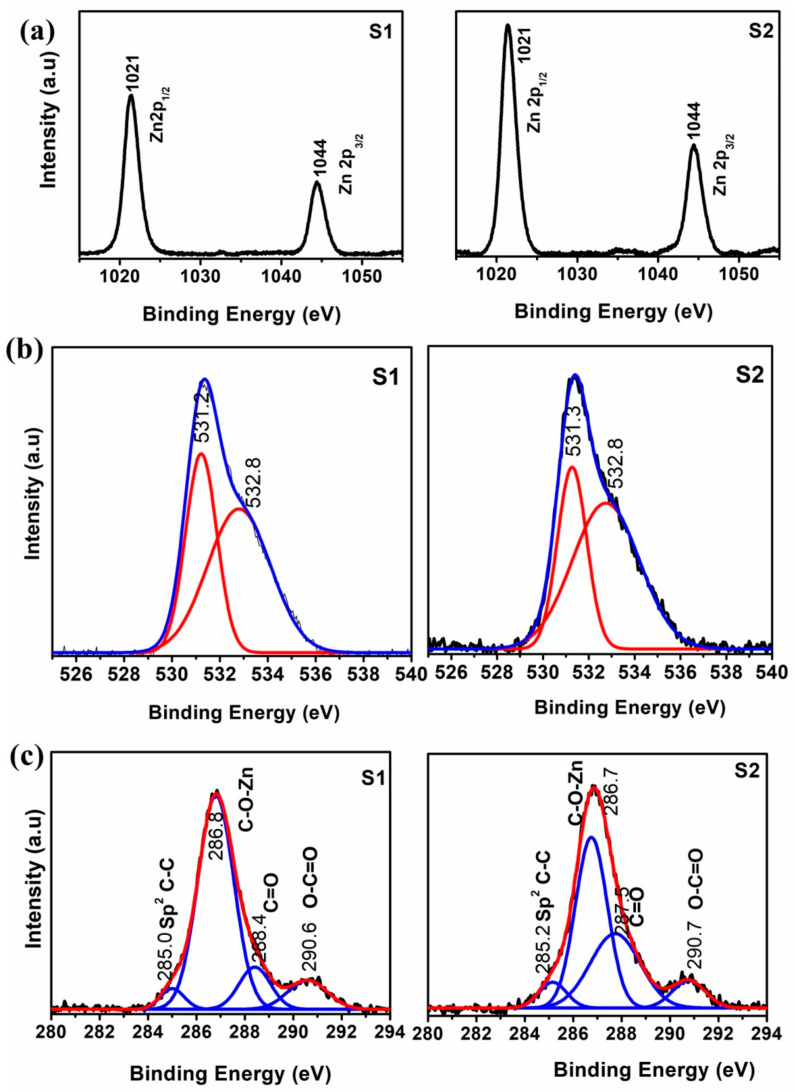
XPS spectrum of (**a**) Zn 2p (**b**) O 1s and (**c**) C 1s peak of the ZnO/graphene nanocomposites.

**Figure 5 micromachines-14-00189-f005:**
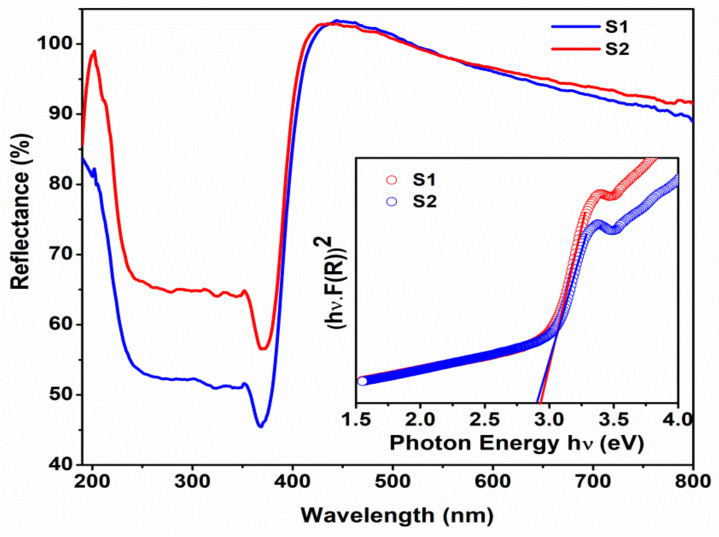
UV-Vis diffused reflectance spectra of ZnO/graphene nanocomposites. (Insert -Tauc plots).

**Figure 6 micromachines-14-00189-f006:**
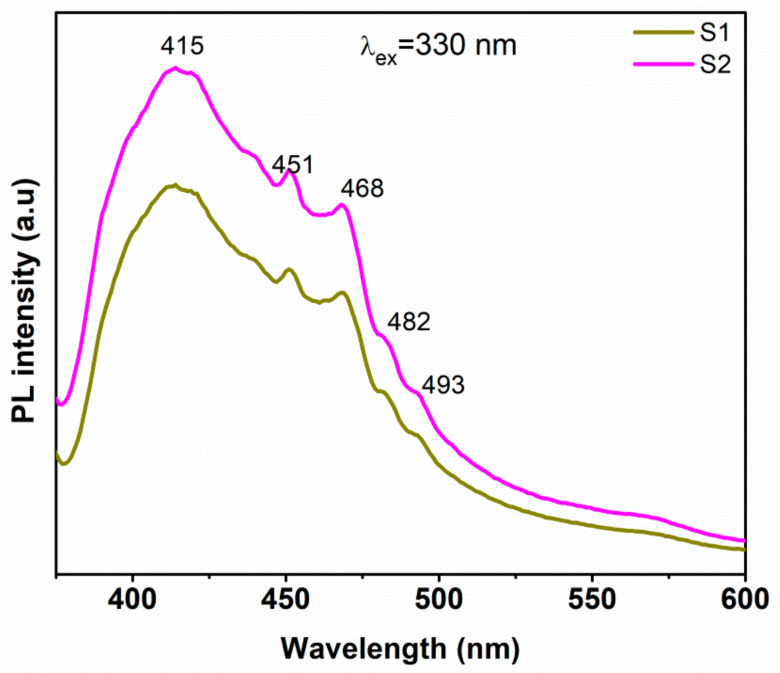
Photoluminescence spectra of ZnO/graphene nanocomposites excited at 330 nm.

**Figure 7 micromachines-14-00189-f007:**
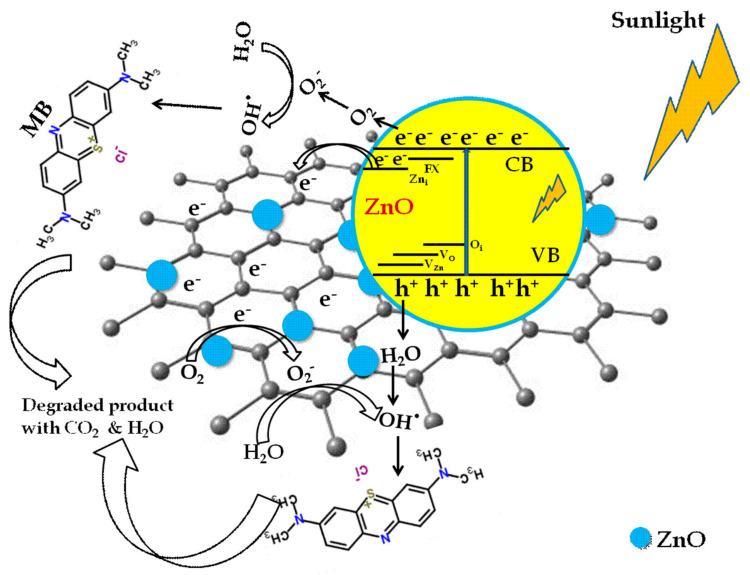
Schematic diagram of photocatalytic degradation mechanism of ZnO/graphene nanocomposite.

**Figure 8 micromachines-14-00189-f008:**
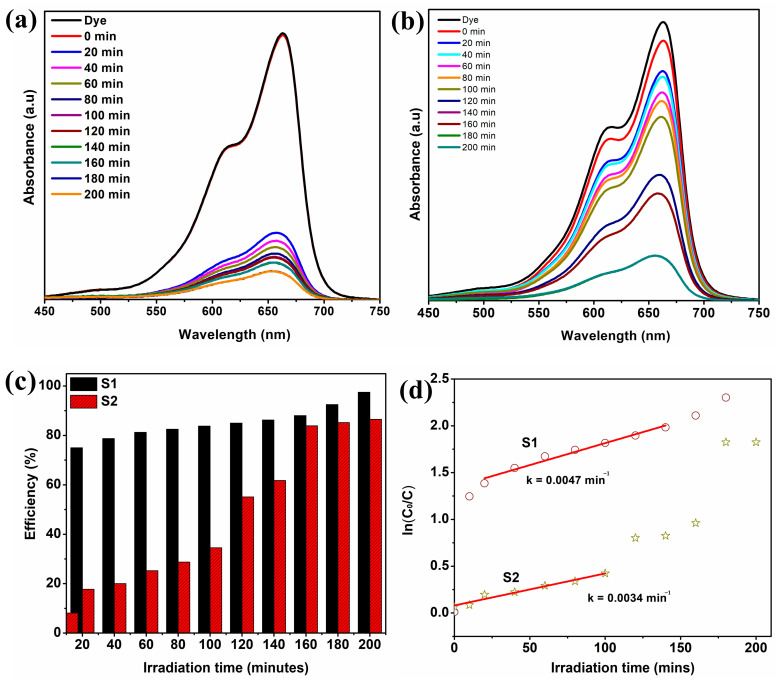
(**a**,**b**) The time-dependent UV-Visible absorption spectra of MB dye solution in the presence of ZnO/graphene, (**c**) Degradation efficiency of ZnO/graphene as a function of irradiation time, and (**d**) Pseudo-first-order kinetic plot ln(C/C_0_) versus irradiation time.

**Figure 9 micromachines-14-00189-f009:**
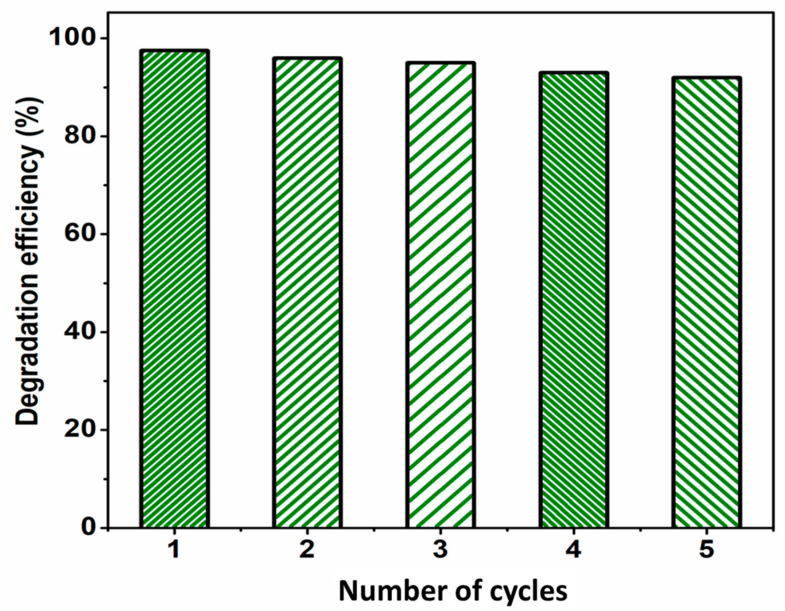
Stability of sample ZnO/graphene nanocomposite (S1) for photocatalytic degradation of MB dye for 5 cycles.

**Table 1 micromachines-14-00189-t001:** Number of layers per graphene flakes, Lateral Size (L), and Crystallite size (L_a_) of ZnO/graphene nanocomposites.

ZnO/Graphene	No. of Layers (n)	No. of Layers (N_G_)	L_a_ (nm)	L (nm)
S1	2	3	70	364
S2	2	3	101	566

**Table 2 micromachines-14-00189-t002:** Comparison of photocatalytic degradation efficiency of ZnO/graphene (S2) with available reports of ZnO/graphene nanocomposites prepared by other methods.

Photocatalyst	Irradiation Light	Degradation Time (min)	Degradation Efficiency (%)	Ref.
ZnO/rGO	Visible	60	99	[68]
ZnO-Graphene	Visible	180	98	[63]
Sn doped ZnO/Graphene	UV	360	96	[69]
ZnO/rGO	UV	120	99	[70]
ZnO/rGO	Visible	180	97	[71]
ZnO-Graphene	Visible	180	99	[30]
ZnO-Graphene	Visible	195	100	[72]
ZnO/GO	UV light	90	98	[73]
ZnO/GO	UV light Visible	120 120	60 38	[74]
ZnO/GO	Solar lamp	90	77.79	[75]
ZnO-Graphene	Sun Light	180	97.5	Present work

## Data Availability

The data presented in this study are available on request from the corresponding author.

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
