# Peer review of "ZnO/Graphene Composite from Solvent-Exfoliated Few-Layer Graphene Nanosheets for Photocatalytic Dye Degradation under Sunlight Irradiation"

_micromachines, 2023, doi:10.3390/mi14010189_

Round 1

Reviewer 1 Report

The manuscript "ZnO/graphene composite from solvent-exfoliated few-layer graphene nanosheets for photocatalytic dye degradation under sunlight irradiation"  is quite interesting. However, this manuscript needs some improvement to be published in the micromachines journal. Here are some improvements that need to be considered:

1-     The author needs to confirm and clarify the novelty of this research. Is it just about the synthesis of ZnO/graphene composite and its application as a catalyst for the degradation of methylene blue, or are there other novelty aspects for this research?

2-     In the XRD section, The authors mentioned that they used the Scherrer formula without adding it. Kindly add it using the following references.

 - El-Sayed, F., Hussien, M. S., AlAbdulaal, T. H., Abdel-Aty, A. H., Zahran, H. Y., Yahia,  I. S., ... & Elhaes, H. (2022). Study of catalytic activity of G-SrO nanoparticles for degradation of cationic and anionic dye and comparative study photocatalytic and electro & photo-electrocatalytic of anionic dye degradation. Journal of Materials Research and Technology, 20, 959-975.

-Hammad, A. H., & Abdel-wahab, M. S. (2022). Photocatalytic activity in nanostructured zinc oxide thin films doped with metallic copper. Physica B: Condensed Matter, 646, 414352.

3-     The authors don’t mention how the bandgap was calculated in the UV-Vis section, and they can use the following reference to calculate it.

-        Yahia, I. S., Jilani, A., Abdel-wahab, M. S., Zahran, H. Y., Ansari, M. S., Al-Ghamdi, A. A., & Hamdy, M. S. (2016). The photocatalytic activity of graphene oxide/Ag3PO4 nano-composite: loading effect. Optik, 127(22), 10746-10757.

4-     Did the authors do the same photocatalytic experiment under sunlight without adding the photocatalyst because it is known that MB dye is affected by light? Kindly provide a figure to confirm it.

5-     Some typos and errors need to be corrected within the manuscript.

6-     The manuscript shows around 30% plagiarism. Kindly see the similarity report.

Author Response

We thank the reviewer for their comments to improve our manuscript. We have done all the suggested corrections. Herewith we are sending the answers to the reviewer’s comments.

Reviewer 1

The manuscript "ZnO/graphene composite from solvent-exfoliated few-layer graphene nanosheets for photocatalytic dye degradation under sunlight irradiation"  is quite interesting. However, this manuscript needs some improvement to be published in the micromachines journal. Here are some improvements that need to be considered:

Thanks for the reviewer comments and suggestions. We have done all the corrections in the revised manuscript as per the reviewer suggestions.

Reviewers comment

1-     The author needs to confirm and clarify the novelty of this research. Is it just about the synthesis of ZnO/graphene composite and its application as a catalyst for the degradation of methylene blue, or are there other novelty aspects for this research?

Reply to the reviewers comment

Thanks for the comment. The solvent used for the exfoliation and the method of ZnO/graphene composite synthesis are the novelty of this work. The introduction section is slightly modified to reflect the novelty of this work

Reviewers comment

2-     In the XRD section, The authors mentioned that they used the Scherrer formula without adding it. Kindly add it using the following references. - El-Sayed, F., Hussien, M. S., AlAbdulaal, T. H., Abdel-Aty, A. H., Zahran, H. Y., Yahia,  I. S., ... & Elhaes, H. (2022). Study of catalytic activity of G-SrO nanoparticles for degradation of cationic and anionic dye and comparative study photocatalytic and electro & photo-electrocatalytic of anionic dye degradation. Journal of Materials Research and Technology, 20, 959-975.‏-Hammad, A. H., & Abdel-wahab, M. S. (2022). Photocatalytic activity in nanostructured zinc oxide thin films doped with metallic copper. Physica B: Condensed Matter, 646, 414352.‏

Reply to the reviewers comment

Thanks for the suggestion. The above mentioned references are added in appropriate place as per the reviewer's suggestion.

Reviewers comment

3-     The authors don’t mention how the bandgap was calculated in the UV-Vis section, and they can use the following reference to calculate it. Yahia, I. S., Jilani, A., Abdel-wahab, M. S., Zahran, H. Y., Ansari, M. S., Al-Ghamdi, A. A., & Hamdy, M. S. (2016). The photocatalytic activity of graphene oxide/Ag3PO4 nano-composite: loading effect. Optik, 127(22), 10746-10757.‏

Reply to the reviewers comment

            Thanks for the suggestion. We have added the suggested reference that explains the procedure to calculate the band gap

Reviewers comment

4-     Did the authors do the same photocatalytic experiment under sunlight without adding the photocatalyst because it is known that MB dye is affected by light? Kindly provide a figure to confirm it.

Reply to the reviewers comment

Thank for the comment. As per the results reported in the following reference, the degradation of MB dye in the acidic/neutral medium under sunlight irradiation without catalyst is negligible.

Soltani, T.; Entezari, M.H. Photolysis and photocatalysis of methylene blue by ferrite bismuth nanoparticles under sunlight irradiation. J. Mol. Catal. A Chem. 2013, 377, 197–203.

However, we have done the experiment. As per the experiment carried out by us for the degradation of dye solution without catalyst, there is no huge degradation is observed after 300 min of sunlight irradiation. The efficiency of degradation after 300 mins is only 8.77%.

Fig. Absorption spectra of dye without catalyst.

Reviewers comment

5-     Some typos and errors need to be corrected within the manuscript

Reply to the reviewers comment

Thanks for the corrections. The typos and errors has been corrected

Reviewers comment

6-     The manuscript shows around 30% plagiarism. Kindly see the similarity report.

Thanks for the comment. We have reduced the similarity. The present similarity is due to the common technical terms which we could not able to avoid.

Reviewer 2

Reviewers comment

This manuscript seems to report the photocatalytic properties of a ZnO and graphene sheets composite.

First of all, the use of sunlight in evaluating photocatalytic properties means that the results cannot be quantitatively quantified. This seems to be a very big mistake in scientific manuscripts. In addition, this manuscript is accompanied by many problems in material analysis and logical explanation. For example, there is no exact description of sample 1 and sample 2. In addition, the authors claim that the graphene peak is not formed in XRD due to the low concentration of graphene and that the increase in ZnO crystals related to the concentration of graphene is not verified, and there are no references. Evaporating the solvent at room temperature to obtain graphene nanosheets is not only very inefficient but also means the presence of many impurities in graphene nanosheet powder.

 In conclusion, this manuscript needs to be overall revised. In particular, not only the overall revision of the experimental design and data analysis is required but also additional analysis such as AFM on the size of the graphene sheet is needed. These tasks are time-consuming. Therefore, the author recommends that the manuscript be withdrawn here, resubmit the manuscript, and resubmit a new manuscript. When revising the manuscript, please refer to the following as well as the above. - The caption of the figure needs a more detailed description of the figure.- Validate the thickness and width of the graphene sheet through AFM.- Compare S1 and S2 for XRD, Raman, and PL analysis.- A minimum of 4 samples is required to account for differences in the concentration of graphene.- Describe the defect analysis by combining the results of XRD, Raman, XPS, and PL.

Reply to the reviewers comment

Thanks for the reviewer comment. However, the use of sunlight in evaluating photocatalytic properties leads to the direct application. Based on the season the sunlight intensity in different region can be quantitatively measured. However the intensity would vary in every year.

The sample description is given in the materials and methods. The ZnO/graphene nanocomposite prepared from 85 and 175 ml of graphene dispersion are named S1 and S2, respectively. We did not observed graphene peaks in XRD. But we confirmed the graphene presence in SEM micrograph and Raman measurements. The reference has been included for the growth of nanoparticles sizes with the presence of graphene. All the figure captions are updated to reflect the sample details. Bothe the samples are compared in XRD, PL and Raman analysis and the defect based discussions were included.

We have done the TEM microscopic measurement for the graphene nanosheets to confirm the few layers and shown in below figure.

Fig. TEM micrograph of graphene nanosheets.

Further the topography of the sheets were evidenced from the AFM microscopy as shown below.

Fig. AFM topographical micrograph of graphene nanosheets.

The size of the graphene nanosheets were discussed in the revised manuscript.

As we have not included the bare ZnO results, the data presented looks qualitatively low. We have included the results of bare ZnO or bare graphene nanosheets to make the manuscript crisper and to deliver the direct outcome. We did not go for higher concentration of graphene for the formation of ZnO/graphene nanocomposites because of our previous experiments showed the decrease of degradation efficiency with higher fraction of graphene in the nanocomposites.

As the S2 sample showed better degradation efficiency, we have done the TEM measurement and the measured TEM micrograph is shown below,

Fig. TEM micrograph of ZnO/graphene nanocomposite (S2).

From the TEM micrograph the ZnO particles on the few layer graphene is evidenced.

We have included the defects based discussion on each results observed in the revised manuscript. The estimated crystallite size of the sample S1 and S2 from XRD is in nanoscale and is greater for sample S2 that that of S1. This indicates the low crystalinity of  S1 than S2 and possibility of the existence of surface defects in both the samples. The same is revealed from the PL result and it evident the presence of  Zn vacancy sites (Vzn), zinc interstitials (Zni), oxygen interstitial (Oi) and oxygen vacancies (Vo) from the emission bands in the visible region. The binding energy peaks at 531.1 and 533.5 eV of XPS results also confirmed the presence oxygen-deficient regions in the samples. The structural defects in the graphene are revealed from Raman spectra which implies the interaction of ZnO with the edge defects of the graphene nanosheets during the composite formation and the defects in the present samples are purely edge defects and are free from basal defects.The obtained band gap of S1 and S2 samples decreases to 2.94 and 2.90 eV compared to that of pure ZnO (3.37 eV). This notable reduction of the band gap of the samples is due to the covalent band created between zinc oxide and graphene.

We hope that we have answered the reviewer comments. We believe that the incorporated reviewer suggestions have improved the quality of the manuscript.

Reviewer 3

The manuscript “ZnO/graphene composite from solvent-exfoliated few-layer graphene nanosheets for photocatalytic dye degradation under sunlight irradiation”, is written well and contains some interesting results. However, I think that this manuscript requires major improvements in following areas:

We sincerely thank the reviewer for the comments and suggestions. We have done all the necessary correction as per the reviewer suggestions.

Reviewers comment

I didn’t observed any quantitative finding in abstract. It must be added.

Reply to the reviewers comment

Thanks for the comment. The abstract has been modified in the revised manuscript.

Reviewers comment

Introduction section has too many paragraph. Combine them with their relevance.

Reply to the reviewers comment

Thanks for the comment. The introduction section has been modified in the revised manuscript.

Reviewers comment

More recent literature work needs to be added.

Reply to the reviewers comment

We have included the recent reports in the revised manuscript.

Reviewers comment

Avoid bulk citations. e.g. Line 45, 56.

Reply to the reviewers comment

Thanks for the comment. As per the suggestions we have removed the bulk citations

Reviewers comment

Add physical and chemical properties of graphene in introduction section. Refer below mentioned articles: https://doi.org/10.1016/j.jmrt.2022.05.076; https://doi.org/10.1016/j.jmrt.2020.05.013

Reply to the reviewers comment

Thanks for the comment. A brief discussion on the importance of physicochemical properties of graphene is included in the revised manuscript and we have also included the suggested references in the revised manuscript.

Reviewers comment

Rewrite all equations as per the journal requirement.

Reply to the reviewers comment

Thanks for the comment. We have tried to keep all the content in the journal format. We changed the equation in the available format in the MDPI template. However, if the publisher want to change any format we are fine with that.

Reviewers comment

Table captions and figure captions are not proper. Arrange them as per the requirement.

Reply to the reviewers comment

Thanks for the comment. The figure and table captions have been corrected.

Reviewers comment

Figure 9: Specify the unit for x axis

Reply to the reviewers comment

Thanks for the correction. The x-axis is don’t have any unit. However, we have changed the name as “Number of cycles” to make it more clear

Reviewers comment

Elaborate this in more detail in discussion section: “The graphene sheets are free from basal plane defects and the existing edge defects facilitate the binding of oxygen with a carbon atom and lead to the formation of C-O-Zn binding”

Reply to the reviewers comment

Thanks for the suggestion. The sentence has been modified in the revised manuscript for the easy understanding.

Reviewers comment

Results and Discussion: Compare your findings with similar studies.

Reply to the reviewers comment

The findings are compared with the similar studies from the available reports and included as table.

Reviewers comment

Add relevant citations in discussion wherever required for the mentioned reasons of your findings.

Reply to the reviewers comment

 The relevant citations have been included in the revised manuscript.

Reviewers comment

Conclusion section need to manage properly in key points.

Reply to the reviewers comment

The conclusion section is modified

Reviewer 2 Report

This manuscript seems to report the photocatalytic properties of a ZnO and graphene sheets composite.

First of all, the use of sunlight in evaluating photocatalytic properties means that the results cannot be quantitatively quantified. This seems to be a very big mistake in scientific manuscripts. In addition, this manuscript is accompanied by many problems in material analysis and logical explanation. For example, there is no exact description of sample 1 and sample 2. In addition, the authors claim that the graphene peak is not formed in XRD due to the low concentration of graphene and that the increase in ZnO crystals related to the concentration of graphene is not verified, and there are no references. Evaporating the solvent at room temperature to obtain graphene nanosheets is not only very inefficient but also means the presence of many impurities in graphene nanosheet powder.

In conclusion, this manuscript needs to be overall revised. In particular, not only the overall revision of the experimental design and data analysis is required but also additional analysis such as AFM on the size of the graphene sheet is needed.

These tasks are time-consuming. Therefore, the author recommends that the manuscript be withdrawn here, resubmit the manuscript, and resubmit a new manuscript.

When revising the manuscript, please refer to the following as well as the above.

- The caption of the figure needs a more detailed description of the figure.

- Validate the thickness and width of the graphene sheet through AFM.

- Compare S1 and S2 for XRD, Raman, and PL analysis.

- A minimum of 4 samples is required to account for differences in the concentration of graphene.

- Describe the defect analysis by combining the results of XRD, Raman, XPS, and PL.

Author Response

We thank the reviewer for their comments to improve our manuscript. We have done all the suggested corrections. Herewith we are sending the answers to the reviewer’s comments.

Reviewer 2

Reviewers comment

This manuscript seems to report the photocatalytic properties of a ZnO and graphene sheets composite.

First of all, the use of sunlight in evaluating photocatalytic properties means that the results cannot be quantitatively quantified. This seems to be a very big mistake in scientific manuscripts. In addition, this manuscript is accompanied by many problems in material analysis and logical explanation. For example, there is no exact description of sample 1 and sample 2. In addition, the authors claim that the graphene peak is not formed in XRD due to the low concentration of graphene and that the increase in ZnO crystals related to the concentration of graphene is not verified, and there are no references. Evaporating the solvent at room temperature to obtain graphene nanosheets is not only very inefficient but also means the presence of many impurities in graphene nanosheet powder.

 In conclusion, this manuscript needs to be overall revised. In particular, not only the overall revision of the experimental design and data analysis is required but also additional analysis such as AFM on the size of the graphene sheet is needed. These tasks are time-consuming. Therefore, the author recommends that the manuscript be withdrawn here, resubmit the manuscript, and resubmit a new manuscript. When revising the manuscript, please refer to the following as well as the above. - The caption of the figure needs a more detailed description of the figure.- Validate the thickness and width of the graphene sheet through AFM.- Compare S1 and S2 for XRD, Raman, and PL analysis.- A minimum of 4 samples is required to account for differences in the concentration of graphene.- Describe the defect analysis by combining the results of XRD, Raman, XPS, and PL.

Reply to the reviewers comment

Thanks for the reviewer comment. However, the use of sunlight in evaluating photocatalytic properties leads to the direct application. Based on the season the sunlight intensity in different region can be quantitatively measured. However the intensity would vary in every year.

The sample description is given in the materials and methods. The ZnO/graphene nanocomposite prepared from 85 and 175 ml of graphene dispersion are named S1 and S2, respectively. We did not observed graphene peaks in XRD. But we confirmed the graphene presence in SEM micrograph and Raman measurements. The reference has been included for the growth of nanoparticles sizes with the presence of graphene. All the figure captions are updated to reflect the sample details. Bothe the samples are compared in XRD, PL and Raman analysis and the defect based discussions were included.

We have done the TEM microscopic measurement for the graphene nanosheets to confirm the few layers and shown in below figure.

Fig. TEM micrograph of graphene nanosheets.

Further the topography of the sheets were evidenced from the AFM microscopy as shown below.

Fig. AFM topographical micrograph of graphene nanosheets.

The size of the graphene nanosheets were discussed in the revised manuscript.

As we have not included the bare ZnO results, the data presented looks qualitatively low. We have included the results of bare ZnO or bare graphene nanosheets to make the manuscript crisper and to deliver the direct outcome. We did not go for higher concentration of graphene for the formation of ZnO/graphene nanocomposites because of our previous experiments showed the decrease of degradation efficiency with higher fraction of graphene in the nanocomposites.

As the S2 sample showed better degradation efficiency, we have done the TEM measurement and the measured TEM micrograph is shown below,

Fig. TEM micrograph of ZnO/graphene nanocomposite (S2).

From the TEM micrograph the ZnO particles on the few layer graphene is evidenced.

We have included the defects based discussion on each results observed in the revised manuscript. The estimated crystallite size of the sample S1 and S2 from XRD is in nanoscale and is greater for sample S2 that that of S1. This indicates the low crystalinity of  S1 than S2 and possibility of the existence of surface defects in both the samples. The same is revealed from the PL result and it evident the presence of  Zn vacancy sites (Vzn), zinc interstitials (Zni), oxygen interstitial (Oi) and oxygen vacancies (Vo) from the emission bands in the visible region. The binding energy peaks at 531.1 and 533.5 eV of XPS results also confirmed the presence oxygen-deficient regions in the samples. The structural defects in the graphene are revealed from Raman spectra which implies the interaction of ZnO with the edge defects of the graphene nanosheets during the composite formation and the defects in the present samples are purely edge defects and are free from basal defects.The obtained band gap of S1 and S2 samples decreases to 2.94 and 2.90 eV compared to that of pure ZnO (3.37 eV). This notable reduction of the band gap of the samples is due to the covalent band created between zinc oxide and graphene.

We hope that we have answered the reviewer's comments. We believe that the incorporated reviewer suggestions have improved the quality of the manuscript.

Reviewer 3 Report

The manuscript “ZnO/graphene composite from solvent-exfoliated few-layer graphene nanosheets for photocatalytic dye degradation under sunlight irradiation”, is written well and contains some interesting results. However, I think that this manuscript requires major improvements in following areas:

  • I didn’t observed any quantitative finding in abstract. It must be added.
  • Introduction section has too many paragraph. Combine them with their relevance.
  • More recent literature work needs to be added.
  • Avoid bulk citations. e.g. Line 45, 56.
  • Add physical and chemical properties of graphene in introduction section. Refer below mentioned articles: https://doi.org/10.1016/j.jmrt.2022.05.076; https://doi.org/10.1016/j.jmrt.2020.05.013
  • Rewrite all equations as per the journal requirement.
  • Table captions and figure captions are not proper. Arrange them as per the requirement.
  • Figure 9: Specify the unit for x axis
  • Elaborate this in more detail in discussion section: “The graphene sheets are free from basal plane defects and the existing edge defects facilitate the binding of oxygen with a carbon atom and lead to the formation of C-O-Zn binding”
  • Results and Discussion: Compare your findings with similar studies.
  • Add relevant citations in discussion wherever required for the mentioned reasons of your findings.
  • Conclusion section need to manage properly in key points.

Author Response

We thank the reviewer for their comments to improve our manuscript. We have done all the suggested corrections. Herewith we are sending the answers to the reviewer’s comments.

Reviewer 3

The manuscript “ZnO/graphene composite from solvent-exfoliated few-layer graphene nanosheets for photocatalytic dye degradation under sunlight irradiation”, is written well and contains some interesting results. However, I think that this manuscript requires major improvements in following areas:

We sincerely thank the reviewer for the comments and suggestions. We have done all the necessary correction as per the reviewer suggestions.

Reviewers comment

I didn’t observed any quantitative finding in abstract. It must be added.

Reply to the reviewers comment

Thanks for the comment. The abstract has been modified in the revised manuscript.

Reviewers comment

Introduction section has too many paragraph. Combine them with their relevance.

Reply to the reviewers comment

Thanks for the comment. The introduction section has been modified in the revised manuscript.

Reviewers comment

More recent literature work needs to be added.

Reply to the reviewers comment

We have included the recent reports in the revised manuscript.

Reviewers comment

Avoid bulk citations. e.g. Line 45, 56.

Reply to the reviewers comment

Thanks for the comment. As per the suggestions we have removed the bulk citations

Reviewers comment

Add physical and chemical properties of graphene in introduction section. Refer below mentioned articles: https://doi.org/10.1016/j.jmrt.2022.05.076; https://doi.org/10.1016/j.jmrt.2020.05.013

Reply to the reviewers comment

Thanks for the comment. A brief discussion on the importance of physicochemical properties of graphene is included in the revised manuscript and we have also included the suggested references in the revised manuscript.

Reviewers comment

Rewrite all equations as per the journal requirement.

Reply to the reviewers comment

Thanks for the comment. We have tried to keep all the content in the journal format. We changed the equation in the available format in the MDPI template. However, if the publisher want to change any format we are fine with that.

Reviewers comment

Table captions and figure captions are not proper. Arrange them as per the requirement.

Reply to the reviewers comment

Thanks for the comment. The figure and table captions have been corrected.

Reviewers comment

Figure 9: Specify the unit for x axis

Reply to the reviewers comment

Thanks for the correction. The x-axis is don’t have any unit. However, we have changed the name as “Number of cycles” to make it more clear

Reviewers comment

Elaborate this in more detail in discussion section: “The graphene sheets are free from basal plane defects and the existing edge defects facilitate the binding of oxygen with a carbon atom and lead to the formation of C-O-Zn binding”

Reply to the reviewers comment

Thanks for the suggestion. The sentence has been modified in the revised manuscript for the easy understanding.

Reviewers comment

Results and Discussion: Compare your findings with similar studies.

Reply to the reviewers comment

The findings are compared with the similar studies from the available reports and included as table.

Reviewers comment

Add relevant citations in discussion wherever required for the mentioned reasons of your findings.

Reply to the reviewers comment

 The relevant citations have been included in the revised manuscript.

Reviewers comment

Conclusion section need to manage properly in key points.

Reply to the reviewers comment

The conclusion section is modified

Round 2

Reviewer 1 Report

The authors have revised the manuscript according to the comments. I feel that the paper can be published in its present form. Thanks

Author Response

We appreciate your Constructive comments that would improve our manuscript

Reviewer 2 Report

Dear Authors, 

I still can't agree with the author's opinion.

For experiments, the data with sunlight are unreliable.

As the author also mentioned in their response, sunlight cannot be quantified because it varies from moment to moment.

This makes the degradation time, degradation efficiency, and pseudo-first-order kinetic plot results unreliable in Table 2, Fig. 8 (c, d) and Fig. 9 in this manuscript.

The number of experimental samples is also too unscientific to explain the phenomenon in two cases. The authors should present the results for a mixture sample of 115 ml and 145 ml of graphene dispersion. 

The author's description of the experimental method is very scarce. For example, the mixing ratio of ZnO and graphene is not specified, and the graphene concentration in the graphene dispersion cannot be found either. In conclusion, it can be seen that the data presented in the manuscript are hardly reliable.

How many hours are supernatant dispersions dried at room temperature?

What is the average size and thickness of graphene in a graphene dispersion?

What is the concentration of the graphene dispersion?

Additionally, the TEM and AFM data shown in the responses were not reflected in the new manuscript.

Author Response

Reply to the reviewer comments

We thank the reviewer for their comments to improve our manuscript. We have done all the suggested corrections and included in the revised manuscript. Herewith we are sending the answers to the reviewer’s comments.

Reviewer 2

I still can't agree with the author's opinion.For experiments, the data with sunlight are unreliable. As the author also mentioned in their response, sunlight cannot be quantified because it varies from moment to moment. This makes the degradation time, degradation efficiency, and pseudo-first-order kinetic plot results unreliable in Table 2, Fig. 8 (c, d) and Fig. 9 in this manuscript.

Thanks for the comments. As the reviewer's said, sunlight cannot be quantified and is not sustainable. But, in the case of photocatalytic degradation of waste water and industrial pollutants in large scale aqueous phase, compare to artificial light sources, sunlight is very cheap, eco-friendly and the efficiency also very close to the photocatalysis by artificial light sources.  We adopted the process from the literature only. There are several reports on the direct utilization of sun light for the photocatalytic dye degradation from different publishers and from different research groups. As the reviewer said, the sun light intensity would vary from moment to moment and region to region. It is known fact. However, the researchers are trying to do the research with available sources. Similarly we have done the experiments with the direct sun light. We are further optimizing the experimental parameters with variation in the sunlight at different months and different places. We have listed a few research articles (including from MDPI) who are all used direct sun light or compared the utilization of direct sun light. Based on the available reports and as per the reviewer suggestions, we have included the location and time of those experiments. The flowing sentences are included in the revised manuscript,

The photocatalysis experiment was performed in a sunny day during summer (April) between 12:00 PM to 3.30 PM (average solar radiation is ~20 MJ/m2) at National institute of Technology Tiruchirappalli, Tamilnadu, India in atmospheric environment.

https://doi.org/10.1007/s12034-020-02291-4

https://doi.org/10.1016/j.jallcom.2021.161169

https://doi.org/10.1016/j.jece.2021.105364

https://doi.org/10.3390/jcs4020049

https://doi.org/10.1016/j.optmat.2021.110854

https://doi.org/10.1016/j.ceramint.2021.08.128

DOI: 10.1039/D0RA10698D

https://doi.org/10.1016/j.envres.2021.111369

The number of experimental samples is also too unscientific to explain the phenomenon in two cases. The authors should present the results for a mixture sample of 115 ml and 145 ml of graphene dispersion. 

Thanks for the suggestions. The reviewer might know that the researchers may not know this kind of suggestion may arise during the review process. Based on the literatures and available facilities we are intended to do some useful research. However, we respect the reviewer comment. Hereafter we would do the experiments with more number of samples and more number of experimental parameters.

The author's description of the experimental method is very scarce. For example, the mixing ratio of ZnO and graphene is not specified, and the graphene concentration in the graphene dispersion cannot be found either. In conclusion, it can be seen that the data presented in the manuscript are hardly reliable.

            The determination of graphene concentration in the dispersion is possible using the UV-Vis absorption spectroscopy data by following the procedure mentioned in supplementary material. As per the data in the present case the ratio of ZnO to graphene is 90:1 for S1 and 45 :1 for S2

How many hours are supernatant dispersions dried at room temperature?

 Supernatant graphene dispersions were dried at room temperature for 48 hours.

What is the average size and thickness of graphene in a graphene dispersion?

The average size of the graphene sheets were ranging from few hundreds of nm to few tens of micrometer which is evidenced from the TEM and AFM micrograph also. The size would be same for the nanocomposites also. The thickness of the graphene sheets might get reduced with the incorporation of ZnO. However we believe that the graphene sheets are in wide range of size distribution.

What is the concentration of the graphene dispersion?

The determination of graphene concentration in the dispersion is possible using the UV-Vis absorption spectroscopy data by following the procedure mentioned in supplementary material. As per the data in the present case the ratio of ZnO to graphene is 90:1 for S1 and 45 :1 for S2

What is the average size and thickness of graphene in a graphene dispersion?

The concentration of graphene in the dispersion calculated from UV-Vis absorption spectrum (given in supplementary Materials) is 124 µg/ml

Additionally, the TEM and AFM data shown in the responses were not reflected in the new manuscript.

The TEM and AFM images were included in the supplementary data of the manuscript.

Reviewer 3 Report

The manuscript has been significantly improved. It can be accepted now in the present form 

Author Response

(The authors gave the same response as above.)

Round 3

Reviewer 2 Report

As the author argues, the direct use of sunlight is valuable as supplementary material. However, it is clear that it cannot be scientific data. However, as suggested by the author, if the solar radiation information of direct sunlight is supported, I think that it can be supported as scientific data to some extent. However, the data must not be arbitrary. The National Statistical Office will provide information on average insolation by region. Please write after checking the contents.

How long will it take to make a graphene dispersion and run the experiment? No matter how hard I think about it, just two samples cannot represent a trend to explain the phenomenon. If you need time, you can ask the editor to give you enough time.

Author Response

I still can't agree with the author's opinion.

For experiments, the data with sunlight are unreliable. As the author also mentioned in their response, sunlight cannot be quantified because it varies from moment to moment. This makes the degradation time, degradation efficiency, and pseudo-first-order kinetic plot results unreliable in Table 2, Fig. 8 (c, d) and Fig. 9 in this manuscript.

Thanks for the comments. As the reviewer's said, sunlight cannot be quantified and is not sustainable. But, in the case of photocatalytic degradation of waste water and industrial pollutants in large scale aqueous phase, compare to artificial light sources, sunlight is very cheap, eco-friendly and the efficiency also very close to the photocatalysis by artificial light sources.  We adopted the process from the literature only. There are several reports on the direct utilization of sun light for the photocatalytic dye degradation from different publishers and from different research groups. As the reviewer said, the sun light intensity would vary from moment to moment and region to region. It is known fact. However, the researchers are trying to do the research with available sources. Similarly we have done the experiments with the direct sun light. We are further optimizing the experimental parameters with variation in the sunlight at different months and different places. We have listed a few research articles (including from MDPI) who are all used direct sun light or compared the utilization of direct sun light. Based on the available reports and as per the reviewer suggestions, we have included the location and time of those experiments. The flowing sentences are included in the revised manuscript,

The photocatalysis experiment was performed in a sunny day during summer (April) between 12:00 PM to 3.30 PM (average solar radiation is ~20 MJ/m2) at National institute of Technology Tiruchirappalli, Tamilnadu, India in atmospheric environment.

https://doi.org/10.1007/s12034-020-02291-4

https://doi.org/10.1016/j.jallcom.2021.161169

https://doi.org/10.1016/j.jece.2021.105364

https://doi.org/10.3390/jcs4020049

https://doi.org/10.1016/j.optmat.2021.110854

https://doi.org/10.1016/j.ceramint.2021.08.128

DOI: 10.1039/D0RA10698D

https://doi.org/10.1016/j.envres.2021.111369

The number of experimental samples is also too unscientific to explain the phenomenon in two cases. The authors should present the results for a mixture sample of 115 ml and 145 ml of graphene dispersion. 

Thanks for the suggestions. The reviewer might know that the researchers may not know this kind of suggestion may arise during the review process. Based on the literatures and available facilities we are intended to do some useful research. However, we respect the reviewer comment. Hereafter we would do the experiments with more number of samples and more number of experimental parameters.

The author's description of the experimental method is very scarce. For example, the mixing ratio of ZnO and graphene is not specified, and the graphene concentration in the graphene dispersion cannot be found either. In conclusion, it can be seen that the data presented in the manuscript are hardly reliable.

 The determination of graphene concentration in the dispersion is possible using the UV-Vis absorption spectroscopy data by following the procedure mentioned in supplementary material. As per the data in the present case the ratio of ZnO to graphene is 90:1 for S1 and 45 :1 for S2

How many hours are supernatant dispersions dried at room temperature?

 Supernatant graphene dispersions were dried at room temperature for 48 hours.

What is the average size and thickness of graphene in a graphene dispersion?

What is the concentration of the graphene dispersion?

The determination of graphene concentration in the dispersion is possible using the UV-Vis absorption spectroscopy data by following the procedure mentioned in supplementary material. As per the data in the present case the ratio of ZnO to graphene is 90:1 for S1 and 45 :1 for S2

What is the average size and thickness of graphene in a graphene dispersion?

The concentration of graphene in the dispersion calculated from UV-Vis absorption spectrum (given in supplementary Materials) is 124 µg/ml

Additionally, the TEM and AFM data shown in the responses were not reflected in the new manuscript.

The TEM and AFM images were included in the supplementary data of the manuscript.
